# Simultaneous Partition Experiment of Divalent Metal Ions between Sphalerite and 1 mol/L (Ni, Mg, Co, Fe, Mn)Cl₂ Aqueous Solution under Supercritical Conditions

Etsuo Uchida *[ID], Keiko Wakamatsu and Naoki Takamatsu

Department of Resources and Environmental Engineering, School of Creative Science and Engineering, Waseda University, Shinjuku, Tokyo 169-8555, Japan; k.waka1128@yahoo.ne.jp (K.W.); takamatunaoki0310@gmail.com (N.T.)
* Correspondence: weuchida@waseda.jp; Tel.: +81-3-5286-3318

**Abstract:** A simultaneous partition experiment of divalent metal ions was performed between sphalerite and 1 mol/L (Ni, Mg, Co, Fe, Mn)Cl₂ aqueous solution under supercritical hydrothermal conditions of 500–800 °C and 100 MPa. The bulk partition coefficient that was defined by $K_{PB}(ZnS) = (x_{MeS}/x_{ZnS})/(m_{Meaq}/m_{Znaq})$ followed the order of Zn $\doteq$ Co $\doteq$ Ni > Fe > Mn > Mg at all temperatures. In the partition coefficient versus ionic radius (PC–IR) diagrams with the logarithmic value of the bulk partition coefficient (log $K_{PB}(ZnS)$) on the vertical axis, and the ionic radius of the six-fold coordinated site on the horizontal axis, Ni shows a positive partition anomaly, and the other elements were almost on the PC–IR curve. Based on the PC–IR curve, the optimum ionic radius for sphalerite existed where the ionic radius was slightly larger than Zn (~0.76 Å). A Ni positive partition anomaly may result from its large electronegativity.

**Keywords:** sphalerite; aqueous chloride solution; partition coefficient; divalent cation; supercritical condition

## 1. Introduction

Elements of ore minerals that were formed in hydrothermal ore deposits were transported by hydrothermal water as a medium, and were precipitated by changes in physicochemical conditions. Information on element partitioning between minerals and hydrothermal water provides important information to estimate the chemical composition of the hydrothermal water that was involved in mineral formation. The hydrothermal water that was involved in the formation of hydrothermal ore deposits can be regarded as an aqueous chloride solution from the fluid inclusion study (e.g., [1]). To clarify the partition behavior of divalent metal ions, such as Ni, Mg, Co, Zn, Fe, and Mn, between minerals and aqueous chloride solutions, the authors conducted element partition experiments under supercritical hydrothermal conditions using sulfide minerals (pyrite, pyrrhotite, and alabandite) [2,3], arsenic sulfide minerals (arsenopyrite and cobaltite) [4], and arsenide minerals (löllingite and sufflorite) [5]. In this study, we conducted partition experiments of the divalent metal ions between aqueous chloride solutions and sphalerite, as the experimental target mineral. Uchida et al. [6] have conducted similar experiments on sphalerite. In the partition experiments on the above-mentioned sulfide minerals, arsenic sulfide minerals, and arsenide minerals, the partitioning of Ni, Mg, Co, Zn, Fe, and Mn was targeted. However, in the experiments of Uchida et al. [6], Mg was not present but Cd was, meaning that it was difficult to compare their results with the above-mentioned experimental results for sulfide minerals, arsenic sulfide minerals, and arsenide minerals. Therefore, in this study, partition experiments of Ni, Mg, Co, Zn, Fe, and Mn were carried out using sphalerite, and a comparison was made with the results of the partition experiments for the above-mentioned minerals. One of the biggest differences between

sphalerite and the minerals mentioned above is that divalent metal ions in sphalerite occupy four-fold coordinated sites, whereas those in the above-mentioned minerals occupy six-fold coordinated sites. One aim of the study is to elucidate the partition behavior of Zn between sphalerite with four-fold coordinated sites and aqueous chloride solution under supercritical hydrothermal conditions.

## 2. Materials and Methods

### 2.1. Sphalerite

Sphalerite is the most important ore mineral of zinc, with an ideal chemical composition of ZnS. Wurtzite occurs as a polymorph at higher temperatures [7]. Sphalerite occurs abundantly in vein-, skarn-, and Kuroko-type ore deposits. Sphalerite with an ideal chemical composition is colorless and transparent. In nature, iron may be dissolved in sphalerite, and sphalerite with a low iron content is brown. When the iron content is high, the mineral becomes opaque and exhibits a dark gray metallic luster. In addition to Fe, natural sphalerite may contain small amounts of manganese, cadmium, indium, gallium, and germanium. Sphalerite is classified as a cubic system, and its crystal often exhibits tetrahedral, octahedral or dodecahedral forms. Zn and S in sphalerite are surrounded by four S and Zn atoms, respectively, and are in four-fold coordinated sites [8].

### 2.2. Solid Starting Material

Zinc sulfide (ZnS) (Lot. No. 70509-30, 99.99%, Rare Metallic Co. Ltd., Tokyo, Japan) solid-phase starting material was used. X-ray powder diffraction analysis identified the solid starting material as sphalerite, and no other substances were present (Figure 1). The width of the X-ray diffraction line was wide, which indicates low crystallinity. Scanning electron microscopy revealed that the particle size was ~2 μm, and that, although unclear, it tended to have a shape close to a tetrahedron (Figure 2).

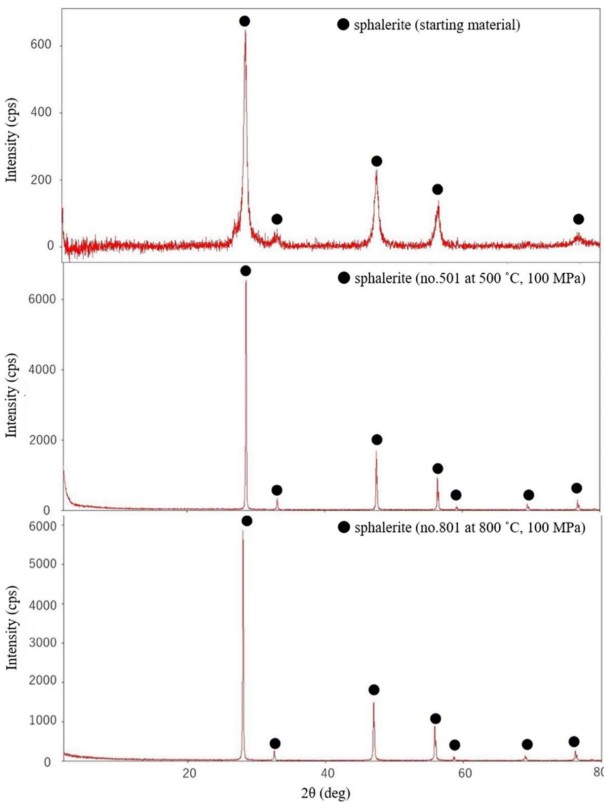

**Figure 1.** Powder X-ray diffraction patterns of starting material (ZnS) and synthesized solid run products.

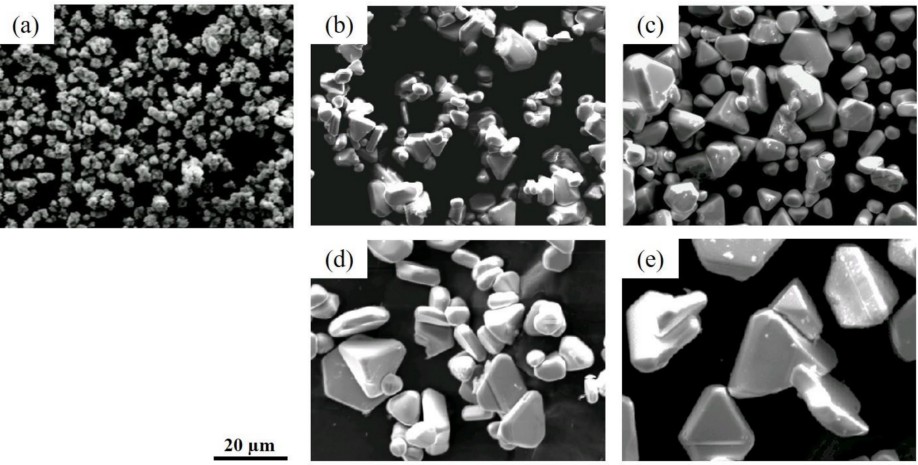

**Figure 2.** SEM images of starting material (ZnS) and solid run products. (**a**) Starting material (ZnS), (**b**) sphalerite synthesized at 500 °C and 100 MPa (run no. 501), (**c**) sphalerite synthesized at 600 °C and 100 MPa (run no. 601), (**d**) sphalerite synthesized at 700 °C and 100 MPa (run no. 701), and (**e**) sphalerite synthesized at 800 °C and 100 MPa (run no. 801).

### 2.3. Liquid Starting Material

As a liquid-phase starting material, a 1 mol/L aqueous chloride solution that was prepared from $NiCl_2 \cdot 6H_2O$, $MgCl_2 \cdot 6H_2O$, $CoCl_2 \cdot 6H_2O$, $FeCl_2 \cdot 4H_2O$, and $MnCl_2 \cdot 4H_2O$ (Junsei Chemical Co., Ltd., Tokyo, Japan) was used. The aqueous chloride solution was mixed in a volume ratio of 1:10:1:1:10 to ensure the analysis of solid-phase products by an energy dispersive X-ray spectrometer and liquid-phase products by an inductively coupled plasma atomic emission spectrometer.

### 2.4. Experimental Procedure

After a carbon electrode was used to weld one side of a gold pipe with a diameter of 3.0 mm, an inner diameter of 2.7 mm, and a length of ~35 mm, 20 to 30 mg of solid-phase starting material and 20 to 30 μL of liquid-phase starting material (Table 1) were added. Approximately 1 mg of anthracene ($C_{14}H_{10}$) (Kanto Chemical Co. Inc., Tokyo, Japan) was added as a reducing agent to keep the iron and manganese in a divalent state. The other end of the gold pipe was welded using a carbon electrode. A gold pipe that contained the starting materials was placed in an oven at 105 °C for ~12 h to check for leakage.

**Table 1.** Experimental conditions in ZnS–(Ni, Mg, Co, Fe, Mn)$Cl_2$–$H_2O$ system.

| Run No. | Duration Days | Temperature (°C) | Pressure (MPa) | Solid (mg) | Fluid (μL) |
|---------|---------------|------------------|----------------|------------|------------|
| 501 | 14 | 500 | 100 | 28.9 | 30 [1] |
| 502 | 14 | 500 | 100 | 30.0 | 30 [1] |
| 503 | 14 | 500 | 100 | 30.0 | 30 [1] |
| 504 | 14 | 500 | 100 | 28.7 | 30 [1] |
| 601 | 7 | 600 | 100 | 30.0 | 30 [1] |
| 602 | 7 | 600 | 100 | 29.9 | 30 [1] |
| 603 | 7 | 600 | 100 | 30.3 | 30 [1] |
| 604 | 7 | 600 | 100 | 30.1 | 30 [1] |
| 701 | 5 | 700 | 100 | 24.5 | 25 [1] |
| 702 | 5 | 700 | 100 | 25.4 | 25 [1] |
| 703 | 5 | 700 | 100 | 25.0 | 25 [1] |
| 704 | 5 | 700 | 100 | 25.5 | 25 [1] |
| 801 | 3 | 800 | 100 | 19.8 | 20 [1] |
| 802 | 3 | 800 | 100 | 20.4 | 20 [1] |
| 803 | 3 | 800 | 100 | 20.4 | 20 [1] |

[1] 1 mol/L$NiCl_2$:$MgCl_2$:$CoCl_2$:$FeCl_2$:$MnCl_2$ = 1:10:1:1:10.

After confirming no leakage, the gold pipe that contained the starting materials was placed in a high-pressure Stellite (Stellite 25) reaction vessel (HP–MRA–114S with 5.5 cm$^3$ volume, HP Technos Co. Ltd., Ibaraki, Japan). The reaction vessel was placed horizontally in an electric furnace (max. 1200 °C, Isuzu Seisakusho Co., Ltd., Tokyo, Japan) to increase the temperature. The temperature was measured using a chromel–alumel thermocouple (Ishikawa Seisakusho, Co. Ltd., Tokyo, Japan) attached to the side of the reaction vessel and was controlled by the temperature controller (DSS83, Shimaden Co. Ltd., Tokyo, Japan). The pressure was adjusted using a high-pressure hand pump (HP-W, max. 200 MPa, Nikkiso Co., Ltd., Tokyo, Japan) through a capillary tube joined to the reaction vessel, using water as the pressure medium. A 200-MPa pressure gauge (Yamazaki Keiki Co., Ltd., Tokyo, Japan) was used to measure the pressure. The reaction temperature was 500–800 °C and the pressure was 100 MPa. The reaction durations were 14 days at 500 °C, 7 days at 600 °C, 5 days at 700 °C, and 3 days at 800 °C. There was no evidence that the gold pipe reacted with the starting materials during the experiments.

After the reaction period was completed, the reaction vessel was removed from the electric furnace and immersed in a stainless-steel bucket that contained water to freeze the reaction. The gold pipe was removed from the reaction vessel when the temperature had dropped sufficiently. The removed gold pipe was weighed to check for leakage.

One side of the removed gold pipe was opened, and the contents were moved into a beaker with a syringe, using pure water. The removed contents were separated into a solid phase and a liquid phase using a Millipore filter (0.45 μm HA). For the solid phase, phase identification was performed using a powder X-ray diffractometer (XRD: RINT-Ultima III, Rigaku, Tokyo, Japan). The solid-phase product was dispersed on a carbon double-sided tape and attached to a glass slide for a rock thin section, and carbon coating (Quick Carbon Coater SC-701C, Sanyu Denshi Co. Ltd., Tokyo, Japan) was conducted. Morphology and particle size were observed using a scanning electron microscope (SEM: JSM-6340, JEOL, Tokyo, Japan). Chemical composition analysis was performed using an energy dispersive X-ray spectrometer (INCA ENERGY, Oxford Instruments, Abingdon, UK) attached to a scanning electron microscope (SEM-EDX). The accelerating voltage was fixed at 15 kV, and the beam current was adjusted so that the X-ray count was 2000 count/s on the Co surface. A chemical composition analysis was performed on the crystal surface. Equilibrium between the mineral surface and aqueous chloride solution was assumed in this study. The analysis time at one point was 1000 s. The analysis was performed on 3 to 5 large crystals with a diameter of 10 μm or more, and the average value was used as the analysis value. Measurements were made on Ni, Mg, Co, Zn, Fe, Mn, and S.

An inductively coupled plasma atomic emission spectrometer (ICP-AES: iCAP6300, Thermo Fisher Scientific Inc., Waltham, MA, USA) was used for the liquid-phase analysis. The analysis was performed on Ni, Mg, Co, Zn, Fe, and Mn.

## 3. Results and Discussion

### 3.1. Solid and Liquid Run Products

XRD indicated that all the solid run products were sphalerite, and no other diffraction lines were observed (Figure 1). Compared with the starting material, the diffraction line sharpened at 500–800 °C, which indicates that the crystallinity was higher than that of the starting material because of recrystallization by high-temperature and high-pressure treatment.

SEM observation indicated that the sphalerite crystals had a tetrahedral shape at any temperature. The diameter increased with an increase in temperature, and reached 2–10 μm at 500 °C, 5–15 μm at 600 °C, 10–20 μm at 700 °C, and 20–30 μm at 800 °C.

SEM-EDX analysis indicated that the Zn content was highest from 98 to 99 mol%. Mg was 0.2–0.7 mol%, Ni was 0.1–0.3 mol%, Co was 0.2–0.6 mol%, Fe was 0.2–0.4 mol%, and Mn was 0.3–1.0 mol% (Table 2). The analysis accuracy (1σ) was 0.25 mol%, 0.05 mol%, 0.11 mol%, 0.10 mol%, 0.09 mol%, and 0.08 mol%, respectively.

Liquid run product analysis by ICP-AES showed that the content of divalent metal ions was 9–19 mol% for Zn, 0.02–0.14 mol% for Ni, 44–52 mol% for Mg, 0.03–0.10 mol% for Co, 1.4–1.8 mol% for Fe, and 33–41 mol% for Mn (Table 2). The analysis accuracy was 0.19 mol%, 0.01 ml%, 0.52 mol%, 0.01 mol%, 0.09 mol%, and 0.41 mol%, respectively.

**Table 2.** Experimental results for ZnS–(Ni, Mg, Co, Fe, Mn)Cl$_2$–H$_2$O system at 500–800 °C and 100 MPa.

| Temp., Pressure | Run No. | | Mg | Mn | Fe | Co | Ni | Zn |
|---|---|---|---|---|---|---|---|---|
| 500 °C, 100 MPa | 501 | Solid [1] | 0.0033 | 0.0033 | 0.0025 | 0.0049 | 0.0014 | 0.9845 |
| | | Fluid (Me$_{aq}$) [2] | 0.64890 | 0.26652 | 0.01031 | 0.00028 | 0.00019 | 0.07380 |
| | | Fluid (MeCl$_{2aq}$) [3] | 0.48141 | 0.16060 | 0.00817 | 0.00020 | 0.00017 | 0.05211 |
| | | log $K_{PB}$ | −3.4149 | −3.0374 | −1.7365 | 0.1143 | −0.2463 | 0.0000 |
| | | log $K_{PN}$ | −3.4363 | −2.9685 | −1.7863 | 0.1143 | −0.3542 | 0.0000 |
| | 502 | Solid [1] | 0.0016 | 0.0034 | 0.0027 | 0.0044 | 0.0016 | 0.9863 |
| | | Fluid (Me$_{aq}$) [2] | 0.65298 | 0.26507 | 0.00939 | 0.00028 | 0.00003 | 0.07226 |
| | | Fluid (MeCl$_{2aq}$) [3] | 0.48566 | 0.15916 | 0.00742 | 0.00019 | 0.00002 | 0.05089 |
| | | log $K_{PB}$ | −3.7471 | −3.0234 | −1.6795 | 0.0724 | 0.6264 | 0.0000 |
| | | log $K_{PN}$ | −3.7708 | −2.9541 | −1.7297 | 0.0724 | 0.5177 | 0.0000 |
| | 503 | Solid [1] | 0.0051 | 0.0030 | 0.0022 | 0.0025 | 0.0008 | 0.9865 |
| | | Fluid (Me$_{aq}$) [2] | 0.65069 | 0.26862 | 0.00993 | 0.00020 | 0.00003 | 0.07053 |
| | | Fluid (MeCl$_{2aq}$) [3] | 0.48307 | 0.16167 | 0.00786 | 0.00014 | 0.00003 | 0.04976 |
| | | log $K_{PB}$ | −3.2555 | −3.1008 | −1.7957 | −0.0429 | 0.2908 | 0.0000 |
| | | log $K_{PN}$ | −3.2776 | −3.0318 | −1.8457 | −0.0429 | 0.1826 | 0.0000 |
| | 504 | Solid [1] | 0.0036 | 0.0037 | 0.0026 | 0.0043 | 0.0018 | 0.9839 |
| | | Fluid (Me$_{aq}$) [2] | 0.64770 | 0.26159 | 0.00918 | 0.00019 | 0.00003 | 0.08131 |
| | | Fluid (MeCl$_{2aq}$) [3] | 0.48044 | 0.15774 | 0.00727 | 0.00013 | 0.00003 | 0.05745 |
| | | log $K_{PB}$ | −3.3356 | −2.9341 | −1.6262 | 0.2773 | 0.7265 | 0.0000 |
| | | log $K_{PN}$ | −3.3567 | −2.8653 | −1.6760 | 0.2773 | 0.6187 | 0.0000 |
| 600 °C, 100 MPa | 601 | Solid [1] | 0.0064 | 0.0072 | 0.0024 | 0.0044 | 0.0028 | 0.9768 |
| | | Fluid (Me$_{aq}$) [2] | 0.45739 | 0.38559 | 0.01699 | 0.00055 | 0.00027 | 0.13922 |
| | | Fluid (MeCl$_{2aq}$) [3] | 0.35369 | 0.30504 | 0.01538 | 0.00047 | 0.00026 | 0.11934 |
| | | log $K_{PB}$ | −2.7015 | −2.5772 | −1.6910 | 0.0630 | 0.1631 | 0.0000 |
| | | log $K_{PN}$ | −2.6568 | −2.5423 | −1.7145 | 0.0630 | 0.1140 | 0.0000 |
| | 602 | Solid [1] | 0.0035 | 0.0077 | 0.0022 | 0.0054 | 0.0030 | 0.9781 |
| | | Fluid (Me$_{aq}$) [2] | 0.4585 | 0.3806 | 0.0152 | 0.0005 | 0.0010 | 0.1442 |
| | | Fluid (MeCl$_{2aq}$) [3] | 0.3550 | 0.3009 | 0.0137 | 0.0004 | 0.0010 | 0.1235 |
| | | Log $K_{PB}$ | −2.9437 | −2.5232 | −1.6733 | 0.1862 | −0.3596 | 0.0000 |
| | | Log $K_{PN}$ | −2.8997 | −2.4882 | −1.6969 | 0.1862 | −0.4089 | 0.0000 |
| | 603 | Solid [1] | 0.0056 | 0.0070 | 0.0020 | 0.0049 | 0.0020 | 0.9786 |
| | | Fluid (Me$_{aq}$) [2] | 0.46019 | 0.37695 | 0.01698 | 0.00056 | 0.00056 | 0.14477 |
| | | Fluid (MeCl$_{2aq}$) [3] | 0.35680 | 0.29769 | 0.01535 | 0.00048 | 0.00053 | 0.12394 |
| | | log $K_{PB}$ | −2.7481 | −2.5620 | −1.7497 | 0.1111 | −0.2742 | 0.0000 |
| | | log $K_{PN}$ | −2.7050 | −2.5269 | −1.7734 | 0.1111 | −0.3237 | 0.0000 |
| | 604 | Solid [1] | 0.0072 | 0.0074 | 0.0041 | 0.0042 | 0.0034 | 0.9737 |
| | | Fluid (Me$_{aq}$) [2] | 0.45951 | 0.38740 | 0.01649 | 0.00049 | 0.00049 | 0.13561 |
| | | Fluid (MeCl$_{2aq}$) [3] | 0.35554 | 0.30612 | 0.01491 | 0.00042 | 0.00047 | 0.11616 |
| | | log $K_{PB}$ | −2.6582 | −2.5767 | −1.4611 | 0.0726 | −0.0222 | 0.0000 |
| | | log $K_{PN}$ | −2.6140 | −2.5417 | −1.4847 | 0.0726 | −0.0716 | 0.0000 |

**Table 2.** *Cont.*

| Temp., Pressure | Run No. | | Mg | Mn | Fe | Co | Ni | Zn |
|---|---|---|---|---|---|---|---|---|
| 700 °C, 100 MPa | 701 | Solid [1] | 0.0037 | 0.0078 | 0.0022 | 0.0054 | 0.0031 | 0.9777 |
| | | Fluid (Me$_{aq}$) [2] | 0.46117 | 0.36207 | 0.01582 | 0.00055 | 0.00018 | 0.16021 |
| | | Fluid (MeCl$_{2aq}$) [3] | 0.42597 | 0.33521 | 0.01533 | 0.00052 | 0.00018 | 0.15250 |
| | | log $K_{PB}$ | −2.8805 | −2.4501 | −1.6513 | 0.2124 | 0.4508 | 0.0000 |
| | | log $K_{PN}$ | −2.8675 | −2.4381 | −1.6591 | 0.2124 | 0.4349 | 0.0000 |
| | 702 | Solid [1] | 0.0020 | 0.0079 | 0.0026 | 0.0050 | 0.0036 | 0.9790 |
| | | Fluid (Me$_{aq}$) [2] | 0.4534 | 0.3623 | 0.0149 | 0.0007 | 0.0002 | 0.1684 |
| | | Fluid (MeCl$_{2aq}$) [3] | 0.4185 | 0.3360 | 0.0144 | 0.0007 | 0.0002 | 0.1605 |
| | | log $K_{PB}$ | −3.1275 | −2.4286 | −1.5241 | 0.0828 | 0.4167 | 0.0000 |
| | | log $K_{PN}$ | −3.1137 | −2.4167 | −1.5317 | 0.0828 | 0.4010 | 0.0000 |
| | 703 | Solid [1] | 0.0022 | 0.0079 | 0.0022 | 0.0044 | 0.0034 | 0.9799 |
| | | Fluid (Me$_{aq}$) [2] | 0.45455 | 0.36658 | 0.01673 | 0.00062 | 0.00031 | 0.16122 |
| | | Fluid (MeCl$_{2aq}$) [3] | 0.41947 | 0.33979 | 0.01622 | 0.00059 | 0.00031 | 0.15358 |
| | | log $K_{PB}$ | −3.1067 | −2.4491 | −1.6676 | 0.0696 | 0.2537 | 0.0000 |
| | | log $K_{PN}$ | −3.0929 | −2.4372 | −1.6753 | 0.0696 | 0.2380 | 0.0000 |
| | 704 | Solid [1] | 0.0035 | 0.0077 | 0.0024 | 0.0056 | 0.0034 | 0.9775 |
| | | Fluid (Me$_{aq}$) [2] | 0.45666 | 0.37653 | 0.01591 | 0.00057 | 0.00043 | 0.14990 |
| | | Fluid (MeCl$_2$) [3] | 0.42119 | 0.34883 | 0.01543 | 0.00054 | 0.00042 | 0.14275 |
| | | log $K_{PB}$ | −2.9347 | −2.5012 | −1.6424 | 0.1775 | 0.0874 | 0.0000 |
| | | log $K_{PN}$ | −2.9208 | −2.4893 | −1.6501 | 0.1775 | 0.0716 | 0.0000 |
| 800 °C, 100 MPa | 801 | Solid [1] | 0.0033 | 0.0103 | 0.0026 | 0.0040 | 0.0015 | 0.9783 |
| | | Fluid (Me$_{aq}$) [2] | 0.51843 | 0.36967 | 0.01830 | 0.00105 | 0.00052 | 0.09203 |
| | | Fluid (MeCl$_{2aq}$) [3] | 0.50382 | 0.35731 | 0.01805 | 0.00102 | 0.00052 | 0.09006 |
| | | log $K_{PB}$ | −3.2287 | −2.5835 | −1.8757 | −0.4398 | −0.5559 | 0.0000 |
| | | log $K_{PN}$ | −3.2257 | −2.5781 | −1.8792 | −0.4398 | −0.5629 | 0.0000 |
| | 802 | Solid [1] | 0.0017 | 0.0101 | 0.0028 | 0.0039 | 0.0031 | 0.9783 |
| | | Fluid (Me$_{aq}$) [2] | 0.46685 | 0.33448 | 0.01578 | 0.00091 | 0.00137 | 0.18061 |
| | | Fluid (MeCl$_{2aq}$) [3] | 0.45317 | 0.32447 | 0.01558 | 0.00090 | 0.00137 | 0.17717 |
| | | log $K_{PB}$ | −3.1634 | −2.2557 | −1.4793 | −0.0999 | −0.3742 | 0.0000 |
| | | log $K_{PN}$ | −3.1589 | −2.2509 | −1.4824 | −0.0999 | −0.3804 | 0.0000 |
| | 803 | Solid [1] | 0.0048 | 0.0088 | 0.0019 | 0.0043 | 0.0026 | 0.9775 |
| | | Fluid (Me$_{aq}$) [2] | 0.46094 | 0.33432 | 0.01588 | 0.00085 | 0.00042 | 0.18759 |
| | | Fluid (MeCl$_{2aq}$) [3] | 0.44731 | 0.32445 | 0.01569 | 0.00083 | 0.00042 | 0.18406 |
| | | log $K_{PB}$ | −2.6965 | −2.2958 | −1.6331 | −0.0069 | 0.0643 | 0.0000 |
| | | log $K_{PN}$ | −2.6917 | −2.2910 | −1.6361 | −0.0069 | 0.0581 | 0.0000 |

[1] Mole fraction of each end member in sphalerite, [2] total molality of the divalent metal ion in the aqueous chloride solution (mol/L), and [3] molality of the neutral species in the aqueous chloride solution (mol/L).

### 3.2. Element Partitioning

The bulk exchange reaction of divalent metal ions between sphalerite and aqueous chloride solution can be written as:

$$ZnS + Me_{aq} = MeS + Zn_{aq} \tag{1}$$

where Me represents a divalent metal ion (Ni, Mg, Co, Fe, and Mn) other than Zn. The subscript aq indicates hydration. The bulk partition coefficient $K_{PB}$ for this reaction can be written as:

$$K_{PB}(ZnS) = \left( \frac{X_{MeS}}{X_{ZnS}} \Big/ \frac{m_{Me_{aq}}}{m_{Zn_{aq}}} \right) \tag{2}$$

where $X_i$ is the mole fraction of each end member in sphalerite, and $m_i$ is the total molality of the divalent metal ion in the aqueous chloride solution. Table 2 shows the logarithmic value of the bulk partition coefficient calculated from each value in Table 2. Figure 3 shows

the temperature dependence of the logarithmic value of the bulk partition coefficient. With the exception of Mn and Mg, the logarithmic value of the bulk partition coefficient shows almost no temperature dependence, and the bulk partition coefficients of Ni, Co and Zn are almost the same and are the largest, followed by Fe, and Mn and Mg, which are the smallest.

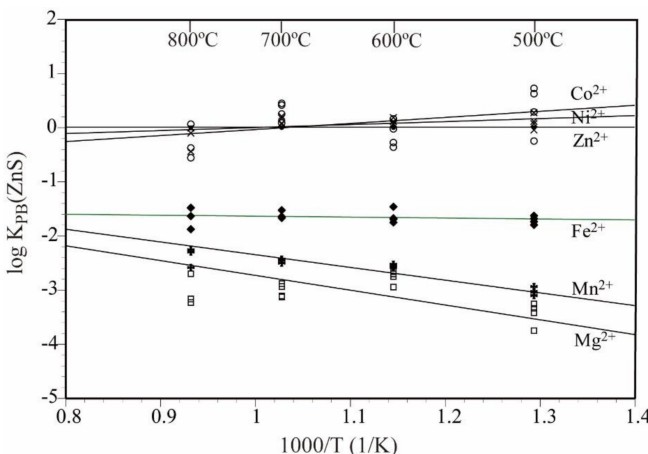

**Figure 3.** Temperature dependence of logarithmic value of bulk partition coefficient (log $K_{PB}$(ZnS)) in ZnS–(Ni, Mg, Co, Fe, Mn)$Cl_2$–$H_2O$ system. The temperature dependence lines were obtained using the least-squares method.

Dissolved species need to be identified to determine the thermodynamically exact partition coefficient. To determine the concentration of dissolved species, it was assumed that only neutral species and tri-chloro complexes formed for Ni, Co, Zn, Fe, and Mn, which are transition metals and easily form chloro complexes, and that no higher-order chloro complex formed besides tri-chloro complexes. It was assumed that $Mg^{2+}_{aq}$, $MgCl^+_{aq}$, and $MgCl_{2aq}$ exist for Mg that easily forms an ionic bond, and that a higher-order chloro complex than the neutral species is not formed. The following mass conservation equations hold between these dissolved species:

$$\Sigma m_{Ni_{aq}} = m_{NiCl^0_{2aq}} + m_{NiCl^-_{3\,aq}} \tag{3}$$

$$\Sigma m_{Co_{aq}} = m_{CoCl^0_2\,aq} + m_{CoCl^-_{3\,aq}} \tag{4}$$

$$\Sigma m_{Zn_{aq}} = m_{ZnCl^0_2\,aq} + m_{ZnCl^-_{3\,aq}} \tag{5}$$

$$\Sigma m_{Fe_{aq}} = m_{FeCl^0_2\,aq} + m_{FeCl^-_{3\,aq}} \tag{6}$$

$$\Sigma m_{Mn_{aq}} = m_{MnCl^0_2\,aq} + m_{MnCl^-_{3\,aq}} \tag{7}$$

$$\Sigma m_{Mg_{aq}} = m_{Mg^{2+}_{\,aq}} + m_{MgCl^+_{\,aq}} + m_{MgCl^0_{2aq}} \tag{8}$$

$$\Sigma m_{Cl_{aq}} = m_{Cl^-\,aq} + m_{MgCl^+_{\,aq}} + 2m_{MgCl^0_2\,aq} + 2m_{NiCl^0_2\,aq} + 2m_{CoCl^0_2\,aq}$$
$$+2m_{ZnCl^0_2\,aq} + 2m_{FeCl^0_2\,aq} + 2m_{MnCl^0_{2aq}} + 3m_{NiCl^-_3\,aq} + 3m_{CoCl^-_3\,aq}$$
$$+3m_{ZnCl^-_3\,aq} + 3m_{FeCl^-_3\,aq} + 3m_{MnCl^-_{3\,aq}} = 2 \tag{9}$$

The following mass action equations hold between each dissolved species:

$$K_{MgCl^+\,aq} = a_{Mg^{2+}_{\,aq}} \cdot a_{Cl^-\,aq} \tag{10}$$

$$K_{MgCl^0_2\,aq} = a_{MgCl^+\,aq} \cdot a_{Cl^-\,aq} \tag{11}$$

$$K_{NiCl^-_3\,aq} = a_{NiCl^0_2\,aq} \cdot a_{Cl^-\,aq} \tag{12}$$

$$K_{CoCl_3^- aq} = a_{CoCl_2^0 aq} \cdot a_{Cl^- aq} \tag{13}$$

$$K_{ZnCl_3^- aq} = a_{ZnCl_2^0 aq} \cdot a_{Cl^- aq} \tag{14}$$

$$K_{FeCl_3^- aq} = a_{FeCl_2^0 aq} \cdot a_{Cl^- aq} \tag{15}$$

$$K_{MnCl_3^- aq} = a_{MnCl_2^0 aq} \cdot a_{Cl^- aq} \tag{16}$$

where $a_{iaq}$ indicates the activity of each dissolved species. The number of dissolved species in Equations (3)–(16) is 14. The molality of all dissolved species can be obtained by solving fourteen equations, that is, seven mass conservation equations (Equations (3)–(9)) and seven mass action equations (Equations (10)–(16)). The formation constants for each dissolved species were from Frantz and Marshall [9], Uchida et al. [10], and Uchida and Tsutsui [11]. Under the experimental conditions, the neutral dissolved species was the most dominant dissolved species (Table 2), and the equilibrium constant $K_{PN}$ when the neutral dissolved species was used is:

$$K_{PN}(ZnS) = \left( \frac{X_{MeS}}{X_{ZnS}} \middle/ \frac{m_{MeCl_2^0 aq}}{m_{ZnCl_2^0 aq}} \right) \tag{17}$$

The logarithmic value of equilibrium constant (log $K_{PN}$(ZnS)), which is defined by Equation (17), is summarized in Table 2. There was almost no difference between the logarithmic value of the bulk equilibrium constant (log $K_{PB}$(ZnS)) and the logarithmic value of the equilibrium constant using neutral dissolved species (log $K_{PN}$(ZnS)).

### 3.3. Partition Behavior of the Elements

Figure 4 shows the partition coefficient versus ionic radius (PC–IR) diagrams using the logarithmic value of the bulk partition coefficient on the vertical axis and the ionic radius [12] at the six-fold coordinated site on the horizontal axis. The PC–IR curves in Figure 4 were obtained with a quadratic equation using the experimental results for Mg, Co, Zn, Fe, and Mn. Almost no change in temperature was observed. The peak of the PC–IR curve is located at a slightly larger ionic radius than Zn, which indicates that Zn and Co can easily enter sphalerite. The element with the smallest partition coefficient is Mg, and the difference between the logarithmic value of the partition coefficient of Zn and Mg is ~3 to 4 digits.

Unlike for pyrite, pyrrhotite [2], alabandite [3], arsenopyrite, cobaltite [4], löllingite, and safflorite [5], Zn was located on the obtained PC–IR curves and showed no negative partition anomaly. It is thought that this occurred because Zn tends to form a four-fold coordination and its partitioning obeys the ionic radius [13]. A similar phenomenon is seen in spinel with four-fold coordinated sites [14].

Ni shows a positive partition anomaly. This tendency is the same as for pyrite, pyrrhotite [2], alabandite [3], arsenopyrite, cobaltite [4], löllingite, and safflorite [5]. This result is explained by the preference of Ni for six-fold coordinated sites [15], but this explanation does not apply for sphalerite with four-fold coordinated sites. The Ni electronegativity is high, which is likely why it easily enters sulfide minerals, arsenic sulfide minerals, and arsenide minerals with high covalent bonds. In spinel, which has the same four-fold coordinated sites and is considered to have strong ionic bonding, Ni does not show a partition anomaly [14].

The position of Fe and Mn in the PC–IR diagrams deviated from the obtained PC–IR curves (Figure 4). The ionic radii used to draw the PC–IR diagrams were obtained from oxides [12]. The deviation of Fe and Mn from the PC–IR curves was commonly observed for sulfide minerals (pyrite, pyrrhotite, and alabandite), arsenic sulfide minerals (arsenopyrite and cobaltite), and arsenide minerals (löllingite and safflorite) (Figure 5 of Uchida et al. [3]). Therefore, we suppose that ionic radii for sulfide minerals, arsenic sulfide minerals, and arsenide minerals differ slightly from those for oxides.

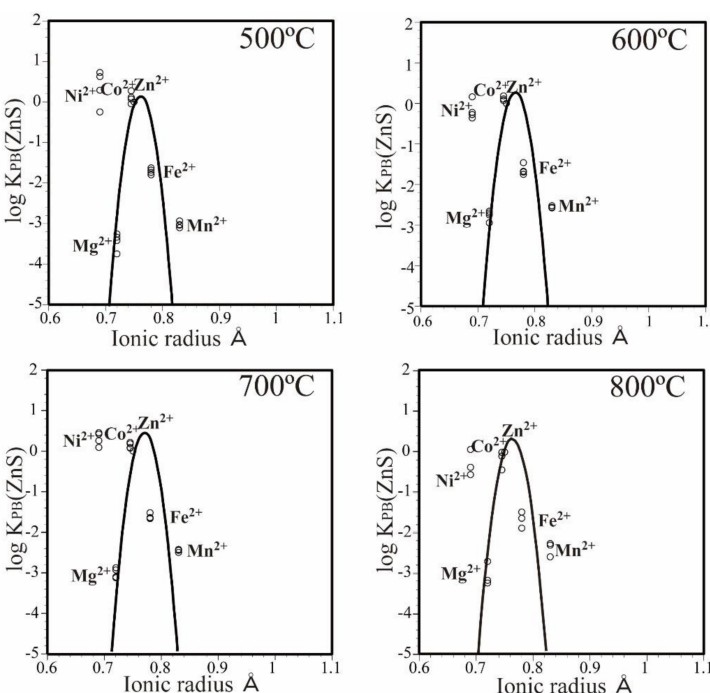

**Figure 4.** Logarithmic value of bulk partition coefficient (log $K_{PB}$(ZnS)) vs. ionic radius (PC–IR) diagrams for the ZnS–(Ni, Mg, Co, Fe, Mn)Cl$_2$–H$_2$O system. PC–IR curves drawn with a quadratic equation and assuming that Mg, Co, Zn, Fe, and Mn will show partition behavior according to their ionic radii.

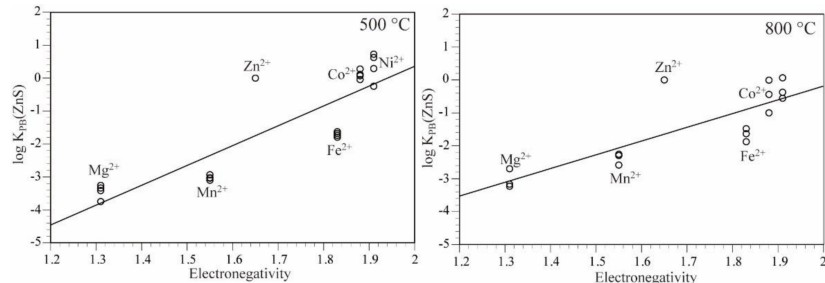

**Figure 5.** Logarithmic value of bulk partition coefficient (log $K_{PB}$) vs. electronegativity diagram for ZnS–(Ni, Mg, Co, Fe, Mn)Cl$_2$–H$_2$O system. Straight lines obtained using a least-squares method and experimental results for Ni, Mg, Co, Fe, and Mn.

Figure 5 shows the logarithmic value of the bulk partition coefficient on the vertical axis and the electronegativity of each element on the horizontal axis [16]. Pauling's electronegativity [17] was used as the electronegativity value. Figure 5 shows a positive proportional relationship between the logarithmic value of the bulk partition coefficient and the electronegativity. Because Ni is plotted on the obtained straight line, the positive partition anomaly was explained by its large electronegativity. Zn deviates significantly from this straight line in the positive direction, which is related to the fact that it is easy to adopt a four-fold coordinated site. However, with respect to Fe, its partition coefficient is small, and cannot be explained even when electronegativity is considered.

### 3.4. Application of Experimental Results to Natural System

Natural sphalerite may contain up to nearly 50 mol% FeS [8] because Fe has a relatively high partition coefficient for sphalerite. In contrast, Mn and Mg are rarely contained in natural sphalerite, which is explained by their small partition coefficients as shown in this experiment. The partition coefficient of Ni and Co with respect to sphalerite is as large as that of Zn, but natural sphalerite contains almost no Ni and Co. This result suggests

that almost no Ni and Co were contained in the hydrothermal water in an environment where sphalerite was formed. Hydrothermal ore deposits that produce sphalerite are often associated with neutral to acidic igneous rocks, which contain small amounts of Ni and Co.

In hydrothermal ore deposits, sphalerite often coexists with pyrite and/or pyrrhotite. Such sphalerite contains a large amount of Fe [18], whereas pyrite and pyrrhotite contain almost no Zn. This is explained by the fact that Zn shows a large negative partition anomaly as shown by the partition experiments of divalent metal ions between pyrite or pyrrhotite and aqueous chloride solutions [2].

## 4. Conclusions

Partition experiments of divalent metal ions were conducted between sphalerite (ZnS) and 1 mol/L (Ni, Mg, Co, Fe, Mn)Cl$_2$ aqueous solution at 500–800 °C and 100 MPa with the following results:

(1)  The bulk partition coefficient of divalent metal ions ($K_{PB}(ZnS) = (x_{MeS}/x_{ZnS}) /(m_{Meaq}/m_{Znaq})$) between sphalerite and aqueous chloride solution followed the order Zn $\fallingdotseq$ Co $\fallingdotseq$ Ni > Fe > Mn > Mg. With the exception of Mg and Mn, almost no temperature dependence was observed for 500–800 °C.

(2)  The optimum ionic radius for sphalerite was slightly larger than the Zn ionic radius (~0.76 Å).

(3)  Because Zn is in the four-fold coordinated site in sphalerite, unlike minerals with a six-fold coordinated site, no negative partition anomaly was observed, and the partition appeared to follow the ionic radius.

(4)  Ni showed a positive partition anomaly in the four-fold coordinated site of sphalerite and in the six-fold coordinated site of sulfide minerals, arsenic sulfide minerals, and arsenide minerals. This is likely because of a large Ni electronegativity.

**Author Contributions:** Conceptualization, E.U.; Data curation, K.W. and N.T.; Funding acquisition, E.U.; Investigation, E.U., K.W. and N.T.; Methodology, E.U.; Project administration, E.U.; Supervision, E.U.; Visualization, E.U., K.W. and N.T.; Writing—original draft, E.U.; Writing—review and editing, E.U.; All authors have read and agreed to the published version of the manuscript.

**Funding:** This research was funded by the Japan Society for the Promotion of Science (JSPS) KAKENHI (Grant No. 19K05356).

**Data Availability Statement:** Data is contained within the article.

**Acknowledgments:** The authors are grateful to the two anonymous reviewers for their insightful reviews and valuable comments to improve the quality of the manuscript. We thank Edanz Group for editing a draft of this manuscript.

**Conflicts of Interest:** The authors declare no conflict of interest.

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
