# Peer review of "Simultaneous Partition Experiment of Divalent Metal Ions between Sphalerite and 1 mol/L (Ni, Mg, Co, Fe, Mn)Cl2 Aqueous Solution under Supercritical Conditions"

_minerals, doi:10.3390/min11040435_

Round 1

Reviewer 1 Report

It was interesting to read this paper, but a have to admit that I cannot recommend it to be published in its current form. The introduction is too short and does not give an up-to-date picture of the addressed problem, nor the significance of described work. The description of the experimental setup is good. Unfortunately, additional analytical studies are required as well as more expanded discussion of results and their processing. If all of this will be revised, it will be a great experimental research. I encourage authors to conduct additional work.

Major questions:

  1. The authors did not provide any information on the behavior of gold in their experiments. Is gold soluble under the given conditions? Are there any sights of gold dissolution or its incorporation into sphalerite?
  2. I wonder if the composition of the crystals’ interior is different from the composition of its surface? In my opinion, the SEM EDS studies of polished material are needed to prove that the equilibrium between crystals and liquid was attained. Alternatively, the bulk composition of crystals may be analyzed by ICP-MS technique to compare with the surface composition measured by SEM EDS. If ions are located on to the surface only, the size of sphalerite crystals must be taken into account.
  3. As far as I’m concerned, at least the thermodynamic model of the ZnS-FeS (ref. below) solid solution is known, as well as the thermodynamics of various forms of Feaq. This makes it possible to calculate the partition coefficient based on literature data and compare it with the result of this work.

Reference:

Martín, J. D., & Gil, A. S. I. (2005). An integrated thermodynamic mixing model for sphalerite geobarometry from 300 to 850 C and up to 1 GPa. Geochimica et cosmochimica acta, 69(4), 995-1006. (This paper contains errors in model parameters, please refer to the electronic appendix or the paper below)

Laptev, Y. V., & Shvarov, Y. V. (2012). Computer simulation in hydrothermal systems with allowance for nonideality of sphalerite and pyrrhotite. Geology of Ore Deposits, 54(4), 304-312.

This thermodynamics of ZnS-FeS solid solution was incroporated into Yuri Shvarovs' HCh software and the partition of Fe between ZnS and liquid can be easily calculated (or this also can be done by hand).

  1. The obtained partition coefficients for nickel and cobalt are quite surprising. They are very high, actually. Previously authors conducted similar studies on pyrite. What are the partition coefficients of Co and Ni for pyrite? What will be the partition of these elements if sphalerite and pyrite were studied simultaneously?
  2. Some of the cited references are in Japanese with English abstract only, and it would be good to provide more information on the result of previous research in the current paper.
  3. The temperature of hydrothermal ore-forming processes is about 150-400C, much lower than the temperature of described experiments. Please, explain how your results can be used to interpret nature processes?

Minor remarks:

  1. No need to plot XRD patterns separately; please redraw into the stacked plot.
  2. In Table 2, units of measurement should be provided.
  3. In Fig.3, does the curve for manganese should be a straight line? It seems like it could be a parabola.

Author Response

To Reviewer 1

We appreciate your insightful comments and suggestions to improve our manuscript. We revised our manuscript taking them into consideration as follows:

Responses to the comments from Reviewer 1

Point 1: The introduction is too short and does not give an up-to-date picture of the addressed problem, nor the significance of described work.

Response: The following sentence was added in “Introduction”: “One aim of the study is to elucidate the partition behavior of Zn between sphalerite with four-fold coordinated sites and aqueous chloride solution under supercritical hydrothermal conditions.”(Line 47-49)

Major questions:

Point 2: The authors did not provide any information on the behavior of gold in their experiments. Is gold soluble under the given conditions? Are there any sights of gold dissolution or its incorporation into sphalerite?

Response: There is no evidence that gold dissolved into aqueous chloride solution under the experimental conditions. In addition, we never detected gold from the solid run products (sphalerite). Thus we added the following sentence in the manuscript: “There was no evidence that the gold pipe reacted with the starting materials during the experiments.”(Line 96-97)

Point 3: I wonder if the composition of the crystals’ interior is different from the composition of its surface? In my opinion, the SEM EDS studies of polished material are needed to prove that the equilibrium between crystals and liquid was attained. Alternatively, the bulk composition of crystals may be analyzed by ICP-MS technique to compare with the surface composition measured by SEM EDS. If ions are located on to the surface only, the size of sphalerite crystals must be taken into account.

Response: We assumed equilibrium between the mineral surface and aqueous chloride solution. Synthesized minerals (run products) usually include solid and liquid inclusions and frequently show chemical zoning. Therefore, we consider that it is better to analyze surface chemical composition of minerals using SEM-EDS rather than bulk chemical composition of solid run products using ICP-MS or ICP-AES. The following sentence was added in the manuscript: “Equilibrium between mineral surface and aqueous chloride solution was assumed in this study.”(Line 116-117)

Point 4: As far as I’m concerned, at least the thermodynamic model of the ZnS-FeS (ref. below) solid solution is known, as well as the thermodynamics of various forms of Feaq. This makes it possible to calculate the partition coefficient based on literature data and compare it with the result of this work.

Reference:

Martín, J. D., & Gil, A. S. I. (2005). An integrated thermodynamic mixing model for sphalerite geobarometry from 300 to 850 C and up to 1 GPa. Geochimica et cosmochimica acta69(4), 995-1006. (This paper contains errors in model parameters, please refer to the electronic appendix or the paper below)

Laptev, Y. V., & Shvarov, Y. V. (2012). Computer simulation in hydrothermal systems with allowance for nonideality of sphalerite and pyrrhotite. Geology of Ore Deposits, 54(4), 304-312.

This thermodynamics of ZnS-FeS solid solution was incroporated into Yuri Shvarovs' HCh software and the partition of Fe between ZnS and liquid can be easily calculated (or this also can be done by hand).

Response: Synthesized sphalerite in this study contained only less than 0.4 mol% FeS. Total content of NiS, MgS, CoS, and MnS is much higher than the FeS content (Table 2). Therefore, it is not possible to discuss ZnS-FeS solid solution in this article.

 We have conducted ion exchange experiments in the ZnS-FeS-(Zn,Fe)Cl2-H2O and ZnS-MnS-(Zn,Mn)Cl2-H2O systems (Kubo, Nakato and Uchida, 1992, Mining Geology) at 600˚C and 100 MPa. Log KPB values obtained by Kubo et al. (1992) were -1.17 and -3.10 for Zn-Fe and Zn-Mn systems, respectively. Log KPB values obtained in this study were -1.64 and -2.56, respectively. The results are more or less consistent to each other.

Point 5: The obtained partition coefficients for nickel and cobalt are quite surprising. They are very high, actually. Previously authors conducted similar studies on pyrite. What are the partition coefficients of Co and Ni for pyrite? What will be the partition of these elements if sphalerite and pyrite were studied simultaneously?

Response: The partition coefficient of Co and Ni against Fe for pyrite is high, that is, log KPB Co or Ni (pyrite) ≑ 0, and that against Zn for sphalerite is also high, that is, log KPB Co or Ni (sphalerite) ≑ 0 (Fig. 5 of Uchida et al., 2020 in Minerals). To know Ni and Co contents in sphalerite and pyrite when they coexist with each other, we need information on Fe/Zn ratio of aqueous chloride solution coexisting with sphalerite and pyrite. Unfortunately, we have no such data. However, as mentioned above, we have conducted ion exchange experiments in the ZnS-FeS-(Zn,Fe)Cl2-H2O system (Kubo et al., 1992). The experimental result showed that the Fe/Zn molar ratio of aqueous chloride solution coexisting with both sphalerite and pyrrhotite was about 10. Therefore, Ni and Co contents in pyrrhotite coexisting with sphalerite is about 10 times higher than those in sphalerite. Similar results are expected for pyrite coexisting with sphalerite.

Point 6: Some of the cited references are in Japanese with English abstract only, and it would be good to provide more information on the result of previous research in the current paper.

Response: Figure 5 of Uchida et al. (2020) in “Minerals” summarized previous experimental results in the form of PC-IR diagrams. Therefore, this figure was referred in the manuscript.(Line 216)

Point 7: The temperature of hydrothermal ore-forming processes is about 150-400C, much lower than the temperature of described experiments. Please, explain how your results can be used to interpret nature processes?

Response: In experiments at lower temperatures than 500˚C, equilibrium between minerals and aqueous chloride solution is difficult to attain, which requires a significantly longer time. This is one reason why we conducted experiments at higher temperatures than 500˚C. Partition coefficient is theoretically proportional to 1/T. Therefore, we can use Arrhenius plot to estimate partition coefficients at lower temperatures (Fig. 3).

Natural hydrothermal water contains a considerable amount of NaCl. In this case, interpretation of the experimental results is complicate. To do so, we need thermochemical data for various aqueous species. Unfortunately, we have not reliable data for the calculation at present.

Minor remarks:

Point 8: No need to plot XRD patterns separately; please redraw into the stacked plot.

Response: Three separated XRD patterns (Fig. 1) were combined into one figure.

Point 9: In Table 2, units of measurement should be provided.

Response: Units of measurement were shown in the footnote of Table 1.

Point 10: In Fig.3, does the curve for manganese should be a straight line? It seems like it could be a parabola.

Response: log K is theoretically proportional to 1/T. Therefore, a straight line is reasonable in Fig. 3.

Reviewer 2 Report

I went through the manuscript entitled "Simultaneous partition experiment of divalent metal ions between sphalerite and 1mol/L (Ni, Mg, Co, Fe, Mn)Cl2 aqueous solution under supercritical conditions" written by Uchida and others. This manuscript deals with the partitioning experiment on sphalerite and divalent metal ions and discusses on the genetic systematics of sulfide minerals. The manuscript is well written and the quality of the experimental data is high enough to be published in an international journal. However, I found some points to be modified. I hope the authors to consider the following points.   1. description of accuracy by two means Section 3.1 shows the chemical analysis of the experimental results. The composition of solid phase is determined using SEM-EDX while that of the liquid is analyzed by ICP-AES. Although two methods determine the composition in different ways, the authors should unify how to describe their accuracy. That is, the result of SEM-EDX is expressed like this: Mn content was 0.3-1.0 mol% and its accuracy (1σ) was 0.08 mol%. In other words, the accuracy is provided as an actual value. On the other hand, the result of ICP-AES is like this: 9-19 mol% for Zn and its accuracy was ~1%. In other words, the accuracy is provided as a relative value. The authors should unify them.   2. partition coefficient vs ionic radius diagrams According to Figure 4, we can easily see that Ni is not on a similar trend to other five elements (Mg, Co, Zn, Fe, Mn). However, the regression line drawn with a quadratic equation does not match the distribution of the points. Other works done by the same authors (such as Uchida et al. 2020 Minerals, cited as 3) show well-fit line. It seems difficult to perform quantitative discussions, those shown in lines 203-209, based on this diagram. The author should provide some explanations that the PC-IR curve does not explain the relationship, especially for Fe and Mn.   Other minor points (mainly editorial) line31: the authors already conducted -> the authors have conducted? line77: μl -> μL (formal expression for liter is L, as the authors did in line68). Similar typo can be seen in Table 1.    

Author Response

To Reviewer 2

We appreciate your insightful comments and suggestions to improve our manuscript. We revised our manuscript taking them into consideration as follows:

Responses to the comments from Reviewer 2

Point 1:  description of accuracy by two means Section 3.1 shows the chemical analysis of the experimental results. The composition of solid phase is determined using SEM-EDX while that of the liquid is analyzed by ICP-AES. Although two methods determine the composition in different ways, the authors should unify how to describe their accuracy. That is, the result of SEM-EDX is expressed like this: Mn content was 0.3-1.0 mol% and its accuracy (1σ) was 0.08 mol%. In other words, the accuracy is provided as an actual value. On the other hand, the result of ICP-AES is like this: 9-19 mol% for Zn and its accuracy was ~1%. In other words, the accuracy is provided as a relative value. The authors should unify them.  

Response: The analysis accuracy was unified to “mol%”.(line 146-147)

Point 2:  partition coefficient vs ionic radius diagrams According to Figure 4, we can easily see that Ni is not on a similar trend to other five elements (Mg, Co, Zn, Fe, Mn). However, the regression line drawn with a quadratic equation does not match the distribution of the points. Other works done by the same authors (such as Uchida et al. 2020 Minerals, cited as 3) show well-fit line. It seems difficult to perform quantitative discussions, those shown in lines 203-209, based on this diagram. The author should provide some explanations that the PC-IR curve does not explain the relationship, especially for Fe and Mn.  

Response: As you pointed out, the position of Fe and Mn in the PC-IR diagrams deviated from the obtained PC-IR curves (Fig. 4). Ionic radii used to draw the PC-IR diagrams were obtained from oxides (Shannon and Prewitt, 1970). The deviation of Fe and Mn from PC-IR curves was observed commonly in sulfide minerals, arsenic sulfide minerals and arsenide minerals, but not in oxide minerals (Fig. 5 of Uchida et al. (2020) in Minerals). Based on this tendency, we suppose that ionic radii for sulfide minerals, arsenic sulfide minerals and arsenide minerals are different more or less from those obtained from oxides. This may be reason for the deviation of Fe and Mn from PC-IR curves. We added this consideration in the manuscript.(Line 212-218)

Point 3: Other minor points (mainly editorial) line31: the authors already conducted -> the authors have conducted?

Response: “the authors already conducted” was replaced with “the authors have conducted”.

Point 4: line77: μl -> μL (formal expression for liter is L, as the authors did in line68). Similar typo can be seen in Table 1.

Response: “μl” was replace with “μL”.

Round 2

Reviewer 1 Report

  • Since "Equilibrium between mineral surface and aqueous chloride solution was assumed in this study.", you should change the title of this manuscript to: "Simultaneous partition experiment of divalent metal ions between the surface of sphalerite...". You studied the surface partition coefficient, not bulk, and this should be clearly stated.
  • Since you analyzed mineral surfaces, it is worth mentioning the working regime of EDS because it affects the depth of the analyzed zone. The contents of the analyzed elements may vary significantly from the crystals' surface towards its core.
  • The solubility of gold in a chlorine-bearing solution can reach up to n*10 ppm (https://doi.org/10.2113/gselements.5.5.281), which too low to be detected by standard SEM EDS technique. I encourage authors to check their samples for the presence of gold in their future research

Author Response

To Reviewer 2

Thank you again for your useful comments and suggestions to our revised manuscript. We treated your comments and suggestions as follows:

Responses to the comments and suggestions from Reviewer 1

Point 1: Since "Equilibrium between mineral surface and aqueous chloride solution was assumed in this study.", you should change the title of this manuscript to: "Simultaneous partition experiment of divalent metal ions between the surface of sphalerite...". You studied the surface partition coefficient, not bulk, and this should be clearly stated.

Response: We don't think it is necessary to add "the surface of" to the title. This is because the equilibrium is certainly established between the surface of the synthesized sphalerite and the aqueous chloride solution, which may not be in equilibrium with the inside of sphalerite. Since this paper deals with the equilibrium between sphalerite and aqueous chloride solution, we don't think it is necessary to add the words "the surface of" to the title.

Point 2: Since you analyzed mineral surfaces, it is worth mentioning the working regime of EDS because it affects the depth of the analyzed zone. The contents of the analyzed elements may vary significantly from the crystals' surface towards its core.

Response: The following sentence was added in the manuscript (line 115-117): “The accelerating voltage was fixed at 15 kV, and the beam current was adjusted so that the X-ray count was 2000 count/s on the Co surface.”

Point 3: The solubility of gold in a chlorine-bearing solution can reach up to n*10 ppm (https://doi.org/10.2113/gselements.5.5.281), which too low to be detected by standard SEM EDS technique. I encourage authors to check their samples for the presence of gold in their future research

Response: Thank you for your information on solubility of gold in an aqueous chloride solution. We will try to check gold concentration in the aqueous chloride solution in the future research.